# eSEE-d: Emotional State Estimation Based on Eye-Tracking Dataset

**DOI:** 10.3390/brainsci13040589

**Published:** 2023-03-30

**Authors:** Vasileios Skaramagkas, Emmanouil Ktistakis, Dimitris Manousos, Eleni Kazantzaki, Nikolaos S. Tachos, Evanthia Tripoliti, Dimitrios I. Fotiadis, Manolis Tsiknakis

**Affiliations:** 1Institute of Computer Science, Foundation for Research and Technology Hellas (FORTH), GR-700 13 Heraklion, Greece; 2Department of Electrical and Computer Engineering, Hellenic Mediterranean University, GR-710 04 Heraklion, Greece; 3Laboratory of Optics and Vision, School of Medicine, University of Crete, GR-710 03 Heraklion, Greece; 4Biomedical Research Institute, Foundation for Research and Technology Hellas (FORTH), GR-451 10 Ioannina, Greece; 5Department of Materials Science and Engineering, Unit of Medical Technology and Intelligent Information Systems, University of Ioannina, GR-451 10 Ioannina, Greece

**Keywords:** emotion classification, arousal, valence, emotion database, neural networks, eye tracking, affective computing

## Abstract

Affective state estimation is a research field that has gained increased attention from the research community in the last decade. Two of the main catalysts for this are the advancement in the data analysis using artificial intelligence and the availability of high-quality video. Unfortunately, benchmarks and public datasets are limited, thus making the development of new methodologies and the implementation of comparative studies essential. The current work presents the eSEE-d database, which is a resource to be used for emotional State Estimation based on Eye-tracking data. Eye movements of 48 participants were recorded as they watched 10 emotion-evoking videos, each of them followed by a neutral video. Participants rated four emotions (tenderness, anger, disgust, sadness) on a scale from 0 to 10, which was later translated in terms of emotional arousal and valence levels. Furthermore, each participant filled three self-assessment questionnaires. An extensive analysis of the participants’ answers to the questionnaires’ self-assessment scores as well as their ratings during the experiments is presented. Moreover, eye and gaze features were extracted from the low-level eye-recorded metrics, and their correlations with the participants’ ratings are investigated. Finally, we take on the challenge to classify arousal and valence levels based solely on eye and gaze features, leading to promising results. In particular, the Deep Multilayer Perceptron (DMLP) network we developed achieved an accuracy of 92% in distinguishing positive valence from non-positive and 81% in distinguishing low arousal from medium arousal. The dataset is made publicly available.

## 1. Introduction

Emotions are psychological, cognitive and behavioral states associated with feelings and thoughts. In the literature, we can find various types of models for the quantification of emotions from classification using the basic emotions such as [1] to coding models based on the facial movements such as the Facial Action Coding System (FACS) [2] and dimensional models. Among the various ways proposed to categorize emotions, researchers have mainly focused on dimensional scales of emotions [3]. Russell’s Circumplex Model of Affect [4] is a two-dimensional space with emotional arousal (EA) and emotional valence (EV) being the two dimensions. EA describes how calming or exciting an emotion is, while EV is the level of pleasantness. The idea behind this two-dimensional space is that emotions are not discrete states but there are correlations, positive or negative, between them. For example, anger is conceptualized as an emotional state with high arousal and negative valence while tenderness is mapped as neutral to low arousal and positive valence. This approach by Russel is significant because it provides a more complex method for classifying emotions than just categorizing them as positive or negative. By incorporating both EA and EV, the model enables the proper description and comprehension of a greater spectrum of emotions.

Recognizing emotions is crucial for interpreting human actions [5]; therefore, the attempt to automatically identify human emotions has led to a whole new, interdisciplinary research area: that of affective computing, which provides new types of human–computer interaction [3,6].

Several types of input information have been used to classify emotional states, such as: facial expressions, voice signals, gestures, physiological responses, heart rate and brain signals [5]. The most often utilized approach is the combination of physiological signals and eye-related measures [7]. At present, non-contact devices, which use infrared cameras, are most commonly used to track ocular movements [8,9,10]. Few researchers, however, have utilized ocular features as the only predictor of emotional arousal and valence levels. These studies attempt to solve either binary [11,12] or multi-class classification problems [13,14,15,16] with success rates for multi-class cases remaining below 80%, whereas binary classification approaches have proven to be more effective, with success rates reaching as high as 93%. Furthermore, neural network-based approaches have been used for the discrimination of various emotional levels. For example, researchers in [5] present a decision tree combined with neural networks for emotional state recognition based on the measures of pupil diameter and gaze positions when viewing emotion-evoking images. Emotion-eliciting images were used as stimulus in [17] toward the development of an emotion recognition system. The findings of prior research revealed that it is possible to recognize a certain emotional state with up to 90% accuracy.

This study has two primary objectives: specifically, (a) to produce a public database with eye-tracking data obtained from emotion-evoking stimuli under a well-defined experimental protocol, which is significantly larger both in terms of participants and raw eye and gaze data distributed than existing datasets, enabling the use of modern AI-based computational methods for the estimation of models that can predict human emotion, and (b) its use in estimating a computationally efficient small feature set which can predict human emotion using a small number of crucial eye-tracking parameters. On the basis of a meta-analysis of previous studies [18], we anticipate that pupil diameter, blink frequency, and fixation time will increase as emotional arousal rises and will therefore be the gaze-retrieved aspects that best predict emotional arousal. Similarly, according to [19,20], pupil diameter and fixation duration are expected to increase in proportion to the negative valence.

A dataset for emotional State Estimation based on Eye-tracking (eSEE-d) combined with self-assessment emotion evaluation is our contribution to the area. The collection includes the eye-tracking data of 48 individuals and their assessments of emotionally intense video clips. The recorded signals contain numerous metrics pertaining to gaze positions, blinks, and pupil features, enabling the extraction and analysis of a large array of eye attributes, including fixations and saccades. In eSEE-d, individuals’ personalities were profiled using a series of self-assessment questionnaires that assess depressive symptomatology, state and trait anxiety, and empathy (see Section 3 for details). Subsequently, each participant engaged in an experiment in which a series of emotion-eliciting video excerpts from popular films where presented to all participants.

The structure of the present manuscript is as follows. Initially, in Section 2, eye features involved in emotional processes are reported, databases containing eye-tracking data for emotion recognition are reported, and their content and limitations are examined. Section 3 presents the experimental scenarios, stimuli selection, annotation explanation and equipment utilized, and an overview of the experimental setup and the methods employed for the assessment of affect and personality traits is outlined. In Section 3.7, the low-level eye metrics analysis and algorithmic process for the extraction of eye and gaze features is explained. Afterwards, in Section 4, statistical correlations between personality and eye features with affective responses are investigated in detail. Additionally, the strategy and structure of a machine learning approach for the identification of emotional states based on neural networks is described and the results from the methods developed for arousal and valence recognition are presented. Furthermore, in Section 5, we interpret our results regarding the statistical and machine learning analysis and perform a benchmarking upon our findings. Finally, the conclusions drawn from the study are recapitulated and discussed.

## 2. Eye-Tracking Databases for Emotion Recognition

Interest in eye movement research dates back to the nineteenth century. Eye-tracking research is applied in areas such as cognitive psychology, neuropsychology, usability testing, or marketing [21]. One of the primary goals of machine learning in affective computing is the ability to recognize users’ emotions for the purpose of emotion engineering. Methods based on electroencephalography (EEG), face image processing, and voice analysis are among the most prevalent techniques for emotion identification. Even while eye-tracking is rapidly becoming one of the most widely used sensor modalities in affective computing, it is still a relatively new method for emotion detection, particularly when employed exclusively.

The establishment of emotional databases that can be linked to multiple modality signals, stimulus materials, and experimental paradigms is crucial to emotion recognition research. Multiple human senses can be stimulated to develop emotions through the use of audio–visual information employed in multisensory media studies. The examination of facial expressions or neuro-physiological signals has been the primary focus of databases for the research of affect recognition based on visual modalities [22,23,24,25,26]. Yet, despite the fact that eye movements have been shown to be valuable indicators of affective response [18], few researchers have concentrated on the creation of relevant databases. The Eye-Tracking Movie Database (ETMD) [27] is a video-oriented database comprised of 10 participants and annotated with continuous arousal and valence ratings. The twelve (12) movie clips about (3–3.5 min) used as stimuli were collected from the COGNIMUSE database [27] to elicit various levels of arousal and valence based on six basic emotions: namely, happiness, sadness, disgust, fear, surprise, and anger. In addition, the dataset is made available to the public and includes eye-tracking parameters pertaining to gaze and fixation positions as well as pupil size. Nevertheless, the aforementioned database lacks blink-related measurements. Moreover, the video clips comprise random dialogue scenes or sometimes a mix of different scenes from the movie, thus distracting the viewer and weakening their evoked emotions.

The EMOtional attention dataset (EMOd) [28] is a diversified collection of 1019 images that trigger a range of emotions with eye-tracking data taken from 16 people in an effort to investigate the relationship between image sentiment and human attention. In addition, the EMOd contains high-level perceptual qualities, such as elicited emotions, as well as intensive image context labels, including object shapes, object attitudes, and object semantic category. Regarding the eye and gaze tracking metrics, the dataset provides fixation sites, duration, and fixation maps, which are accessible to the public. However, no raw eye and gaze data are provided, thus introducing limitations regarding the management of data for different purposes.

In addition, the datasets 208 NUSEF [29] and CAT2000 [30] are somewhat useful. NUSEF is a collection of 751 emotive photographs, primarily depicting faces, nudity, and human movement. The CAT2000 training set contains 2000 images depicting various settings, including emotive imagery and cartoons. Yet, these two datasets lack emotional content and object labels. In addition, the eye-tracking data acquired from the participants in these two studies is not available to the general public.

Despite their substantial contribution to the scientific community, the databases described have certain limitations. The primary restriction is the limited number of participants and available eye and gaze metrics. This is especially crucial when examining relationships between eye movements and emotions, as some measures, such as blinks, are useful indications of emotional arousal [18]. A second limitation of the datasets mentioned is the fact that they are appropriate for studying the influence of emotional charge on the shifting of attention specifically to the visual stimuli based on fixation saliency maps, but they are not appropriate for studying the correlation of emotional states with gaze patterns and pupil characteristics. Finally, the datasets suffer from a lack of available eye and gaze metrics, which in turn restrains the variety of computational and algorithmic apporaches that can be performed.

The limitations of each of the aforementioned datasets demonstrate the need for the development of a new eye-tracking dataset intended for emotion recognition. To this goal, the eSEE-d database presented in this work is a one-of-a-kind resource that can support new eye-tracking analysis for emotion identification research. It is, to the best of our knowledge, the first publicly accessible eye-tracking based dataset that integrates eye and gaze movements signals with self-assessment of the users while viewing 10 emotion-eliciting video clips (duration: 1–2 min.), enabling the evaluation of the effect and relationship of eye movements with emotion and personality. eSEE-d is also the largest eye-tracking database in terms of the number of participants and the quantity of eye and gaze measurements, thus providing the opportunity for many different types of experimentation both in terms of data management and the development of a variety of machine and deep learning techniques.

## 3. Methods

In this section, we detail the methodology used to generate the dataset and the materials utilized in the research.

### 3.1. Participants

The experimental protocol (110/12-02-2021) was submitted and approved by the Ethical Committee of the Foundation for Research and Technology Hellas (FORTH).

There were a total of fifty-six (56) participants in the study. Seven were eliminated because they did not fulfill the inclusion criteria: six had CES-D scores above the threshold and one had binocular visual acuity worse than 0.10 logMAR. One participant was disqualified due to poor quality recordings. All analyses were conducted on the remaining forty-eight (48) subjects (27 female, 21 male). Their average binocular visual acuity at 80 cm was −0.11 ± 0.08 logMAR (range: 0.10 to (−0.29) logMAR), their mean age was 32 ± 8 years (range: 18–47 years), and their average education level was 17 ± 2 years (range: 12–21 years).

Exclusion criteria for all subjects included any known ocular disease, spectacle-corrected binocular visual acuity in 80 cm of less than 0.10 logMAR (0.8 decimal acuity equivalent), clinically significant aberrant phorias, any known cardiovascular disease, and a CES-D score of 19 or higher (see Section 3.4).

### 3.2. Video Set

In this work, we decided to use videos as emotion-evoking stimuli. Although picture-based elicitation approaches have certain advantages, such as being simple to acquire materials for and constructing the experimental paradigm, they are incapable of providing adequate continuous stimulation, resulting in suboptimal emotion induction [25]. In contrast, a video-based elicitation method can evoke emotion continuously due to prolonged stimulus. Hence, the video-based elicitation method can compensate for the loss of picture stimulation, which has gained considerable attention [31,32].

Ten (10) videos with sound were used in the study, which were obtained from the public database FilmStim [33] and modified to meet the needs of the study. They were cropped so that their duration was shorter than 2 minutes, and no important dialogues were included, since the participants were native Greek speakers and the videos were in English or French.

The dataset consisted of two videos for each of the four chosen emotions (anger, disgust, sadness and tenderness) and two more videos which served as emotionally neutral ones. Table 1 shows the videos used, their duration and their emotion annotation. According to Russell’s arousal–valence space (Figure 1), each emotion corresponds to its level of arousal and valence. Anger and disgust are High Arousal–Negative Valence (HANV), sadness is Low Arousal–Negative Valence (LANV) and tenderness is Low Arousal–Positive Valence (LAPV). Neutral is Medium Arousal–Medium Valence (MAMV). The first three emotions were chosen because they are commonly considered to be amongst the basic emotions [34], they are widely studied in emotion research [35] and their essence is easy to understand [33]. Tenderness, on the other hand, is not considered to be one of the basic emotions, but it has widely been used during the last years in emotion research [36,37,38]. Tenderness and amusement were the only positive emotions in the FilmStim database instead of the more generic “happiness”. Tenderness is also an attachment-related emotion, and thus, it belongs to another, underrepresented group of emotions. Finally, it is an emotion that can easily be evoked by films [33].

In addition, ten (10) neutral videos were used after each emotion-evoking video, with the objective of inducing a relaxing state prior to the next emotion-evoking video and in parallel to allow for the previously evoked emotion to fade away. The neutral videos had a duration of approximately 60 seconds, which enabled us to sample the induced emotion before a user’s emotional response returns to zero or to a baseline level for the first time (see Section 1 for details).

### 3.3. Emotion Annotations

The videos used were already annotated regarding the emotion that they evoke by [33]. The emotions were corresponded to the emotional arousal and valence levels of the valence–arousal space by the study group. This annotation will be referred to as “Objective annotation” from now on.

On the other hand, the “Subjective annotation” was based on each participant’s self-assessment after each video. The self-assessment consisted of a 4-word differential emotions scale (DES)—anger, disgust, sadness and tenderness—on an 11-point scale.

Similar to [39], only self-assessment scores equal to or higher than 4 were accepted as a significant indication of the presence of a specific emotion. A self-assessment score lower than 4 was treated as emotionally neutral. An emotion was selected as “Subjective annotation” if it received a higher rating (at least 1 point) than the other three emotions [37].

### 3.4. Demographics and Psychoemotional Scales

All participants were asked to fill the following set of questionnaires in this order:*Clinical cardiovascular record:* a questionnaire that consisted of two questions: whether the participant had a cardiovascular record and whether he or she is taking any related medication.*CES-D scale:* a short self-report scale designed to measure depressive symptomatology [40,41]. It consists of 20 items that request the participant rate how often over the past week they experienced symptoms associated with depression, such as restless sleep, poor appetite, and feeling lonely. A score greater than 19 was used as the cutoff score in order to identify individuals at risk for clinical depression.*Demographics:* questionnaire with basic demographic information (gender, age, education level).*STAI-trait test:* the State-Trait Anxiety Inventory (trait) is a commonly used measure of trait anxiety [42,43]. It consists of 20 items that are rated on a 4-point scale. It evaluates the anxiety of the participant during the last six months.*Basic Empathy Scale:* questionnaire consisting of 20 items measuring emotional and cognitive empathy [44,45].*STAI-state test:* the State-Trait Anxiety Inventory (state) is a commonly used measure of state anxiety [42,43]. It consists of 20 items that are rated on a 4-point scale. It evaluates the anxiety that the participant feels at the moment of the assessment.

### 3.5. Materials and Setup

The videos were presented on a computer screen (DELL, 24″, 1280 × 720) at 80 cm distance from the participant as shown in Figure 2. Wireless headphones were used, and the sound volume was set. However, participants were asked whether they were comfortable with the sound volume, and it was adjusted if necessary.

Using the Pupil Labs “Pupil Core” eye-tracker, eye-tracking measures were captured [46]. The binocular recordings had a sample rate of 240 Hz, an accuracy of 0.60 deg., and a precision of 0.02 deg. To minimize head movements, all measures were taken with the subjects seated in a chair with their heads supported by a chin and head rest.

Using European-wide standardized logMAR charts, standardized logMAR acuity was determined [47]. With the cover test, stereopsis was assessed.

With the room lights on, controlled photopic lighting conditions were created for recording purposes. The corneal illuminance was 400 lux when the screen was off and 450 lux when the screen was blank.

### 3.6. Experimental Procedure

All participants read and signed an Informative Consent Form before the trial. The participants were then escorted to the laboratory. Subsequently, tests of binocular visual acuity at 80 cm and stereopsis were administered.

Subsequently, participants were asked to complete the above-mentioned questionnaires on the computer screen.

In case there was a cardiovascular record or a CES-D score over 19 (risk for clinical depression), the procedure ended and the participant was excluded. Otherwise, the procedure continued.

Following the questionnaire part, the experimental part commenced. After the gaze-tracker calibration, the guidelines were presented and the participants were informed that they could stop the video if they did not feel comfortable, at any time they wanted, just by pressing the Space button. Next, the emotion-evoking videos were presented in a randomized order. After each emotion-evoking video, participants were presented with a neutral video of about 1 min so the evoked emotion faded away before the next emotion-evoking video. After the neutral video, the questionnaire for the emotion self-assessment was presented to the participants. We purposely decided to introduce a one-minute period before affective annotation from individuals who rated their feelings in response to the previous emotion-evoking clip under a self-assessment protocol targeting to simulate real world scenarios. This time period enables us to sample the emotion before a user’s emotional response returns to zero or to a baseline level for the first time. The design of the study is shown in Figure 3.

For the course of the video-viewing procedure, a member of the research team monitored the gaze-tracker’s output on a second monitor in case of any anomalies in the recordings or the participant needed additional assistance.

To safeguard the participants and the study team from the SARS-CoV-2 pandemic and to prevent the spread of the virus, every precaution was taken.

### 3.7. Data Analysis Methodology

In this subsection, the algorithm used to calculate and interpret eye and gaze-related features from the raw data captured by the gaze-tracking device is described.

#### 3.7.1. Raw Eye-Tracking Data

The raw gaze points from the pupil core are processed and analyzed for each recording sequence to confirm that participants viewed the entire video scene each time and did not glance away from the screen or even closed their eyes for a period longer than the average blink time to avoid viewing. Specifically, we excluded every recording during which the participant closed his/her eyes for a duration longer than 10% of the total clip duration to avoid watching, as blinking causes around 5–10% data loss during a recording [48]. Additionally, we considered faulty recordings those in which participants’ attention was decoupled from the screen stimuli (mind-wandering phenomenon) for a length greater than 20% of the overall duration of the corresponding video clip [49]. The 20% cutoff was selected based on the evidence provided by a range of studies indicating that mind wandering occurs at least 20% and up to 50% of the time, even during tasks that are not designed to induce it [50,51]. Additionally, if a participant abruptly paused the video clip, the relevant recording was deleted. As a result, the final dataset consisted of 476 recordings that include those obtained for each of the valid emotion-evoking videos watched by the 48 study participants.

The output of the gaze tracker used in this study includes the gaze positions (x, y coordinates), the blink timings (start and end times), and the pupil diameter in millimeters. These metrics comprise various sorts of noise originating from both the eye tracker and the participants, as it is widely known that when collecting gaze-related data, there is typically some noise owing to eye blinking and an inability to capture corneal reflections [52]. Thus, filtering and denoising must be applied to the eye movement data in order to eliminate this undesired variance.

The raw gaze coordinates, which are in the form of normalized pixels, are converted into degrees of visual angle, and the instantaneous sample-to-sample gaze movement between two consecutive gaze points is calculated, leading to the calculation of the angular velocity, given the sampling frequency $F_{s}$. To reduce velocity noise, we utilized a 5-tap velocity filter provided by [53] after one large velocity peak value (i.e., during a saccadic movement).

#### 3.7.2. Fixation and Saccade Detection

Fixation detection algorithms categorize gaze data according to dispersion, velocity, and acceleration (or combinations thereof) parameters; [53,54]). Fixations and saccades are recognized based on the Velocity-Threshold Identification (I-VT) algorithm proposed by [54] due to its superiority when sample-by-sample comparisons are taken into account [55]. Furthermore, to determine the length of the fixations, we added an extra minimum duration threshold. The steps of the algorithm are as follows:Compute point-to-point velocities for every protocol point.Mark every point below the velocity threshold as a fixation point and every other location as a saccade point.Collapse consecutive fixation points based on fixation time into fixation groups, omitting saccade points.Map each cluster of fixations to the fixation at the centroid of their points.Return fixations.

In the I-VT algorithm, the velocity threshold for saccade detection was set to 45 deg./s, as in [55]. In addition, the minimum fixation duration threshold was determined at 55 ms [56].

#### 3.7.3. Pupil and Blink Detection

The pupil diameter and blink timings determined by the eye tracker contribute to the extraction of additional information linked to the pupil and blink. In the infrared illuminated eye camera frame, the pupil recognition algorithm locates the dark pupil [46]. The algorithm is not affected by corneal reflection and may be used by those who use contact lenses or spectacles. Based on input from user-submitted eye camera footage, the pupil recognition algorithm is constantly being improved. Based on a confidence threshold corresponding to the effective detection of the pupil area, blinks start and end times are determined. In total, 28 eye and gaze features are extracted based on fixation, saccade, blink and pupil characteristics and are presented in Table 2.

Fixation duration (median, variation coef., skewness, kurtosis);Fixation frequency;Saccade duration (median, variation coef., skewness, kurtosis);Saccade velocity (mean, variation coef., skewness, kurtosis);Peak saccade velocity (mean, variation coef., skewness, kurtosis);Saccade amplitude (median, variation coef., skewness, kurtosis);Saccade frequency;Blink duration (mean);Blink frequency;Pupil diameter (mean, variation coef., skewness, kurtosis).

#### 3.7.4. Pupil Estimation Implications

The effect of emotional arousal and valence on pupil size is complex due to the fact that pupil diameter and its variation are highly reliant on various factors, including lighting conditions [5,57], the luminance of the movie [14] and the adapting field size [58,59]. There have been attempts to remove the movie luminance effect on pupil diameter [60]. Recent research involves deriving the estimated pupil diameter from the measured pupil diameter using the V component of the HSV color space [14].

In the present work, the experimental setup was designed to minimize this effect as much as feasible. First, the room’s lighting settings were adjusted to be photopic so that any brightness variations in the film would be insignificant. Second, for the same reason, the participant’s distance from the screen was quite large (80 cm). Corneal illuminance during films was measured and varied between 390 and 411 lux, showing little variance among the films.

To determine if the influence of the film’s luminance was sufficiently low, a linear regression analysis was performed between pupil diameter and the V component of the HSV color space for each video. Among the 48 participants, 24 showed very weak correlation (r < 0.20), 20 showed weak correlation (0.20 < r < 0.40) and 4 showed moderate correlation (0.41 < r < 0.47). The levels of correlation were set based on [61].

It was considered that the results of linear regression analysis were satisfying, since no strong correlation between pupil diameter and V component was found in any of the participants. Thus, all analysis on pupil diameter was carried out with that acquired from the gaze tracker pupil diameter.

## 4. Results

In this section, the study’s findings are provided. First, a statistical study is presented, which is followed by a Machine Learning analysis to detect any correlation between ocular characteristics and arousal and valence levels.

### 4.1. Statistical Analysis

#### 4.1.1. Questionnaire Analysis

Descriptive statistics of the five self-assessment scales are presented in Table 3. An independent sample *t*-test showed only an emotional empathy score was statistically significantly different between men and women (t(47) = 3.538, *p* = 0.001) with women showing a higher score by 0.58. No correlation was found between any of the scales and age or education level (*p* > 0.092).

#### 4.1.2. Objective and Subjective Emotion Annotation Analysis

Table 4 shows the Hit Rate and the Mean Rating for each target emotion. Hit Rate is the percentage of the videos for which the participants had indicated that they had felt the target emotion (objective annotation) at least one point more intensely than any of the other three untargeted emotions [39]. Mean Rating is the mean value of the scores for each target emotion. Its scale is from 0 to 10, as decided based on methodological and theoretical criteria [62].

The relation between objective and subjective arousal is presented in Figure 4, where the subjective annotations are displayed with reference to the objective ones. Each pie chart shows the distribution of the subjective annotations corresponding to the target emotion class of the valence–arousal space. For this analysis, the valence–arousal space is divided into four quadrants, and the emotions are placed in the space according to its arousal and valence levels. The quadrants are “High Arousal–Positive Valence (HAPV)”, “High Arousal–Negative Valence (HANV)”, “Low Arousal–Negative Valence (LANV)” and “Low Arousal–Positive Valence (LAPV)”. The origin of the system is “Medium Arousal–Medium Valence”.

The videos with objective annotation HANV, i.e., anger and disgust, are subjectively annotated as: (a) HANV in a percentage of 77%, (b) MAMV, i.e., neutral, in 12% and (c) LANV in 10%. The ones with objective annotation LANV, i.e., sadness, are subjectively annotated as: (a) LANV in 71%, (b) MAMV in 21% and (c) HANV in 7%. The subjective annotation of the videos with objective annotation “tenderness” (LAPV) are annotated as: (a) LAPV in 71% and (b) MAMV in 27%. The neutral videos received a subjective annotation as neutral (MAMV) in a percentage of 85%. These results reveal that in a substantial percentage, the videos did evoke to participants the target emotion.

Based on [39], we evaluated the “Discreteness” of the videos, i.e., whether the rating of the emotion that was identified as “subjective annotation” was statistically significant greater than the ratings of the rest of emotions. We used a *t*-test to make a pairwise comparisons between the subjective annotation and each of the rest of the emotions. In all comparisons, the rating of the emotion that was identified as “subjective annotation” was statistically significant greater than the ratings of the rest of the emotions (*p* < 0.001). Table 5 shows the mean ratings of each emotion in each subjective annotation.

#### 4.1.3. Eye Feature Analysis

As there are several samples of the same subject at the same level of arousal or valence, our dataset is comprised of dependent observations. Hence, an analysis based on the assumption of independent observations within each group or between the groups themselves, such as ANOVA, is rejected. In order to discover which ocular characteristics are affected by the arousal and valence levels, a Mixed Linear Model (MLM) evaluation was conducted on each feature individually. Mixed Linear Models are an extension of simple linear models that are used to evaluate both fixed and random effects. MLMs provide more accurate estimates of the effects, better statistical power and non-inflated Type I errors compared to traditional analyses [63]. In the MLM analysis, arousal or valence level was selected as the fixed component, while participant ID was selected as the random factor. In addition, a Bonferroni post hoc test was performed to compare class characteristics..

In Table 6 and Table 7, only the features that were affected in a statistically significant manner by the arousal and valence levels are presented. The Bonferroni post hoc test showed that five features were statistically significantly different between low and medium arousal level, six features were different between low and high, and nine features were different between medium and high. As far as the valence level is concerned, seven features were statistically significantly different between negative and medium valence level, three features were different between negative and positive, and six were different between medium and positive.

In order to evaluate the simultaneous effect of the arousal and valence levels on each eye feature, the subjective emotion class of the valence–arousal space was selected as fixed factor in the Mixed Linear Model (MLM) analysis. Again, a Bonferroni post hoc test for evaluating features among the classes was performed. Table 8 shows the features that were affected in a statistically significant manner by the emotion class. One feature was statistically significantly different between LAPV and HANV, six features were statistically significantly different between MAMV and HANV and between MAMV and LAPV, five were statistically significantly different between LANV and HANV, two were statistically significantly different between LANV and MAMV, and four were statistically significantly different between LANV and LAPV.

Estimated mean values of the Mixed Linear Model and standard errors of all eye features for three classes (low, medium, high) of emotional arousal and three classes of emotional valence (negative, medium, positive) are presented in Table 2.

When the self-assessment scores were added to the Mixed Linear Model, no statistically significant effect of them on any of the eye features was found.

### 4.2. Feature Selection Process

Even though deep learning-based models enable a feature extraction process, it might be a good idea to remove irrelevant features before training the model. This may reduce memory and time consumption since deep learning procedures usually require a large amount of data. Moreover, feature selection could enhance the ability of the model to learn the most and least significant features and thus exclude it from the future data collection, resulting in improved performance [64].

For the machine learning analysis presented in Section 4.3, we developed a model including specific eye-tracking features for the estimation of the levels of arousal and valence as presented in Section 4.1. To this goal, we created three different feature sets for our predictive models. Specifically, we produced a correlation matrix of the features that were statistically significantly different among the arousal and valence levels and the emotion classes (Table 6 and Table 7), and from the pairs that were highly correlated (r > 0.3), we kept the features that better distinguish among the levels of arousal and valence. Thus, for the estimation of the levels of EA, the feature set includes the metrics of Fixation and Blink frequency, Pupil diameter, Fixation duration kurtosis, Saccade duration variation and PD variation. For the identification of EV levels, our feature set comprises of eight eye-tracking metrics: Fixation and Blink frequency, Saccade amplitude and duration, Pupil diameter, Fixation duration kurtosis, Saccade duration variation and Pupil diameter kurtosis. Finally and for the synchronous estimation of EA and EV levels, the selected feature set includes the metrics of Saccade and Blink frequency, Saccade amplitude, Pupil diameter, Fixation duration kurtosis, Saccade duration variation and Pupil diameter kurtosis.

### 4.3. Machine Learning Analysis

This section examines the relationships between fixation, saccade, blink, and pupil-related ocular characteristics with arousal and valence levels. Consequently, the EV outcome measure can take on three values: negative, neutral, or positive, while the EA response variables are set to high, medium, and low. Specifically, the divided EA instances **low, medium** and **high** are marked as class **LA, MA** and **HA**, respectively. In the same manner, **negative, neutral** and **positive** EV examples are denoted as classes **NV, MV** and **PV**, respectively.

In our work, we used a set of DMLP neural networks written in the Python 3.8 environment for our models and generated a partition with 80% of the samples for training the networks and the remaining samples for testing, according to the Pareto principle [65]. Figure 5 illustrates the basic structure of our neural networks. The first layer contains Nϵ{6,7,8,28} neurons based on the size of the data vector per experiment followed by *m* hidden layers, where mϵZ∩[2,3]. Additionally, each hidden layer contains *n* neurons, where nϵZ∩[8,256]. All layers are followed by a dropout layer with a 0.25 rate and Rectified Linear Unit (ReLu) as their Activation Function due to its computational efficiency [66]. Moreover, for the last layer, we used the Sigmoid function for the binary classification procedure and the Softmax function for the multi-class problems [67]. The optimizer used to accelerate the learning process is “Adam” with binary cross-entropy as the loss function. Each network is characterized by a single output *y*.

In this study, several neural network topologies were compared. Several numbers of hidden neurons have been utilized for training and testing in order to discover the optimal network layout for each classification attempt. The network with the least number of hidden neurons and the smallest testing error has been determined to be the optimal one. The resulting neural network with reduced complexity has been trained and evaluated again. We next conducted a 10-fold cross-validation and evaluated the performance of the neural networks using the average f1-score, Area Under Curve (AUC), and accuracy rate of well-classified testing data. The achieved results for the superior neural network architectures are presented in Table 9, Table 10 and Table 11. Table 9 represents the results of identifying various levels of EA, Table 10 presents the results of the discrimination attempt between different EV levels, while Table 11 provide the synchronous identification attempt of arousal and valence levels.

From Table 9, we observe that the selected neural network is capable of discriminating the presence of a high emotional arousal level to an accuracy of 74%. However, without the presence of the medium class, it is more challenging for the neural network to discern between high and low arousal. Such behavior is not observed when HA is separated from MA or when MA is separated from LA. Additionally, especially with reference to the multi-class categorization of the three classes HA, MA, and LA, the network’s percentage of right predictions is maintained at 74% by adding a third hidden layer.

Regarding the objective of identifying and categorizing the different levels of emotional valence, the PV class is sufficiently distinct from the MV and NV classes. Simultaneously, class PV is easily differentiated from NV and MV at a rate of 82% and 90%, respectively. In comparison, NV is deemed to be weakly discriminated from MV. Despite the addition of an extra hidden layer, it appears to affect the success rate of predictions when categorizing the three classes.

An additional approach was examined in order to investigate the potential of creating a combined model to synchronously identify the levels of EA and EV. Based on the annotation given by the participants, we split the response variable into four classes depending on the combined arousal and valence levels. The respective classes, the structure of the network as well as the results are presented in Table 11.

It is worth noting that a different method was explored in relation to the models constructed and discussed in the preceding paragraphs. This strategy entailed the inclusion of the scores from the five questionnaires in the input layer of the networks. We executed the neural networks by adding in the input layer of the network each of the five questionnaires’ scores separately and in aggregate to determine their effect on the models’ performance. The results indicated that adding the scores had no discernible effect on the model’s efficiency, either increasing or decreasing.

## 5. Discussion

In this work, we presented an eye-tracking dataset for the assessment of emotional arousal and valence levels based simply on eye and gaze characteristics. The dataset consists of eye and gaze-recording signals from 48 participants who viewed 10 emotionally evocative videos. The participants scored each emotional clip according to four basic emotions, which were then translated into arousal and valence levels.

Our statistical research revealed that numerous characteristics can be used to distinguish between arousal and valence levels (Table 6 and Table 7). Based on the current research, we anticipated that pupil diameter, blink frequency, and fixation length would be the characteristics that best predict arousal and valence levels. This hypothesis was validated, although other variables, such as fixation frequency and saccade amplitude, contributed considerably to our prediction model. As hypothesized, our attempt to discriminate emotional states revealed that High Arousal–Negative Valence (HANV) and Low Arousal–Positive Valence (LAPV) could be distinguished with excellent accuracy. In addition to this comparison, additional characteristics such as fixation frequency, fixation duration, fixation duration kurtosis, saccade frequency, saccade amplitude, saccade duration variation and skewness, blink frequency, and pupil diameter can be used to distinguish HANV from other emotional states. Low Arousal–Negative Valence (LANV) can be discriminated from other emotional states by saccade amplitude, saccade velocity skewness and kurtosis, saccade duration, and pupil diameter (Table 8). In general, it appears that pupil diameter, blink frequency, fixation frequency, and saccade amplitude were strongly influenced by arousal, valence, and emotional states. These results demonstrate the ability of the eye-tracking data collected during the designed experimental protocol to indicate the level of EA and EV.

Regarding our neural network analysis, the highest success rate was observed during the binary classification between not positive and positive emotional valence level, achieving 92% accuracy whereas PV is effectively distinguished among the rest of the two levels. However, integrating the “neutral” class proved challenging, resulting in a significant decrease in network performance, especially when differentiating NV and MV. By adding an additional hidden layer to the shallow neural network model, the neural network was still able to reliably predict the three levels of EV despite a decline in performance. In terms of forecasting emotional arousal levels, both binary and multi-class identification tests provided positive results, with up to 81% accurate predictions; nevertheless, the LA class had a significant impact on the networks’ performance. Positive results demonstrated that a model is able to forecast synchronously the amounts of EA and EV with a 72% success rate.

Overall, we tested a wide variety of eye features related to fixations, saccades, blinks and pupil and investigated the potential of several neural network models on different classification scenarios. In agreement with prior research, our findings reveal the effect of emotional charge on eye movements and pupillary responses. Specifically and in line with the ideas of [11,12,16], high arousal and positive valence levels can be successfully identified based solely on eye-tracking features. Furthermore, when comparing our work to those of [13,14,15], our results demonstrate significant improvements in terms of accuracy which can be attributed in part to the significant larger size of the training database. In addition, we have verified that using neural networks for emotion level estimation produces encouraging results, superior to [5,17], thus indicating potential toward the development of an emotion identification system with high discretization ability.

Although the videos used in this study were obtained from a public database, they are extracted from well-known film productions. Hence, it could be argued that the fact some participants being familiar with a certain video could result in participants not being as emotionally charged as they would have been upon first viewing. One more potential limitation of our study is that the participants rated the videos based on the five eliciting emotions rather than rating directly the levels arousal and valence, which were set later in accordance with the emotion ratings. Moreover, the inclusion of tenderness as a target emotion at the expense of the basic emotion of happiness means that the set did not contain any high intensity positive emotions and thus may result in a relative imbalance. However, tenderness and amusement were the only positive valence emotions in the FilmStim database. Furthermore, it is important to take into consideration that evoking high intensity–positive valence emotions to the subjects within a lab setting, and solely based on movie clip content, is extremely difficult and would probably lead to eliciting neutral valence emotions labeled wrongfully as high valence ones (i.e., happiness) and thus diminishing the integrity of eSEE-d.

We used Hit Rate as a variable that indicates the relation between the objective and the subjective annotation, and we saw that only 53% of the videos that were objectively annotated as anger were also subjectively annotated as anger. Another 20% was annotated as sadness and another 12% was annotated as disgust. This result confirms a rather old finding that anger is difficult to discretely elicit with brief films because it tends to co-exist with other negative emotions [39]. This limitation does not directly affect our further results because all analysis was performed having the subjective annotation as a basis and not the objective one. This co-existence of emotions, however, may have an impact on the subjective annotation as well. The DES self-assessment questionnaire gives the participants the choice to rate more than one emotion, but we defined subjective annotation as the emotion that received a higher rating (at least 1 point) than the other three emotions. In order to evaluate our definition of subjective annotation, we performed pairwise comparisons between the subjective annotation and each of the rest of the emotions. The t-test shows that although there is co-existence of the negative emotions, the rating of the emotion that was set as subjective annotation was statistically significantly greater that the rest of the emotions and confirms that the subjective annotation reflects mainly the corresponding emotion.

The FilmStim database consists of both colored and black and white videos. For the present study, videos from both categories were selected. Pupil size and pupil responses are known to be greatly dependent on luminance contrast, and this is why the study setup was such that luminance effect would be minimized, and this is confirmed in Section 3.7.4. Although there are studies that have shown that pupil size depends on color, apart from luminance [68], it is accepted that pupillary responses are affected mainly by luminance contrast and not color contrast [69]. Thus, since it is shown that luminance effect was not significant in our study, color effect would be even lower. It would be interesting though to have an exact evaluation of this effect in our study.

A final issue that needs to be pointed out is the fact that a there was a 1-min time interval between the emotion-evoking video and the self-assessment questionnaire. This approach was selected to imitate as effectively as feasible a real-world setting in which a person would be unable to appraise their emotions immediately after their manifestation. Emotions are dynamic, complicated processes that develop over time. So, a complete theoretical knowledge of how they operate cannot be attained until their time-related characteristics are well understood [70]. A recent review article [71] presents research on the determinants of emotion duration, building on a previous preliminary review [72]. One central temporal characteristic of emotions is the duration of emotional experience, which has been defined as the amount of time that elapses between the beginning and end point of an emotional episode. In contrast to moods, emotions begin with the occurrence of an external or internal event [73], despite the fact that the beginning of the feeling does not always correspond with the commencement of the event. An emotional episode concludes when the intensity of the emotional response returns to zero or to a baseline level, either for the first time [74], for several consecutive times [75], or permanently [73].

Despite the fact that duration definitions and associated metrics vary across accessible studies, the first definition (i.e., initial return to zero or baseline) is most frequently employed. Regardless of the time conceptualization adopted, it has been discovered that duration is highly diverse, with emotions ranging from a few seconds to several hours or even longer. In this sense, not all feelings are created equal: some emotions have been discovered to last for a long time, while others tend to go away rapidly. However, according to modern neurology, the average duration of an emotion in the human brain is 90 s, and therefore, we assume that the 1-min time does not affect the self-assessment.

## 6. Conclusions

In this study, we presented a dataset consisting of eye movements recorded while each subject watched emotion-eliciting films and evaluated five emotions, which were then converted into emotional arousal and valence levels. Despite the significant contribution to advancing the relevant research to date of the available public databases, the eSEE-d dataset consists of a significantly larger number of participants, and it includes a plethora of eye and gaze metrics. Thus, it provides the necessary depth and breadth of information for both experimenting with novel machine learning methods in attempting to develop optimal models for predicting emotional state using eye-tracking features.

The results of this study indicate the potential of neural networks for differentiating between different emotional states and emphasize the urgent need for further research. In this approach, we are currently investigating new models as well as the application of advanced deep learning methods in order to develop a model for simultaneously assessing valence and arousal levels with high efficiency. In addition, we intend to experiment with alternate, more robust feature selection methods as opposed to the statistical methods provided in this paper. Deep feature extraction architectures will be utilized for this task, and a different data-handling approach will be tested in order to predict the broad variety of different emotional states. Lastly, we intend to compare our findings with those of other studies that apply multimodal techniques, i.e., employ additional biosignals, and study the necessity and potential of merging eye and gaze data with other biometrics for improved performance in terms of computing expense.

This dataset is made accessible to the public, and we strongly encourage other researchers and academics to test their methods and algorithmic approaches on this immensely challenging database.

## Figures and Tables

**Figure 1 brainsci-13-00589-f001:**
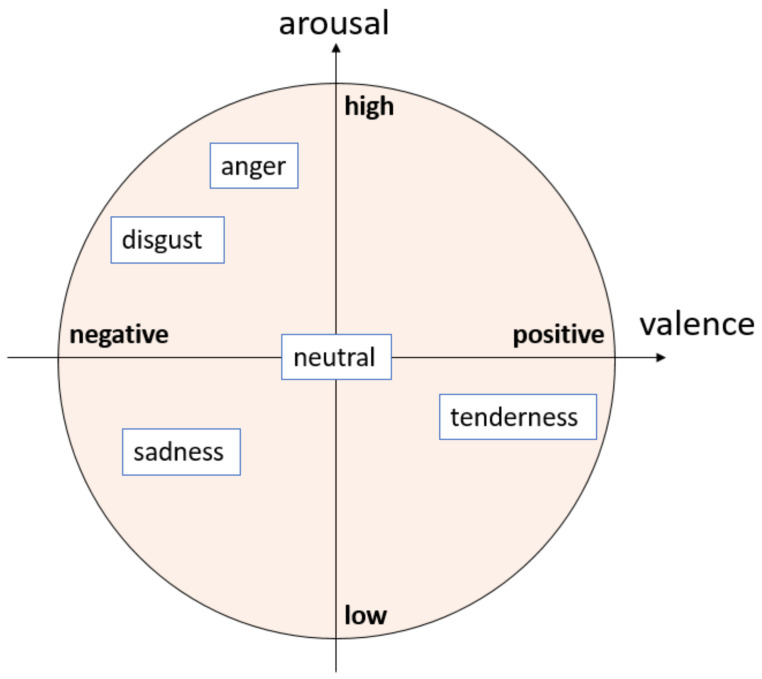
Valence and arousal space with the corresponding location of the emotional classes that were used in the study.

**Figure 2 brainsci-13-00589-f002:**
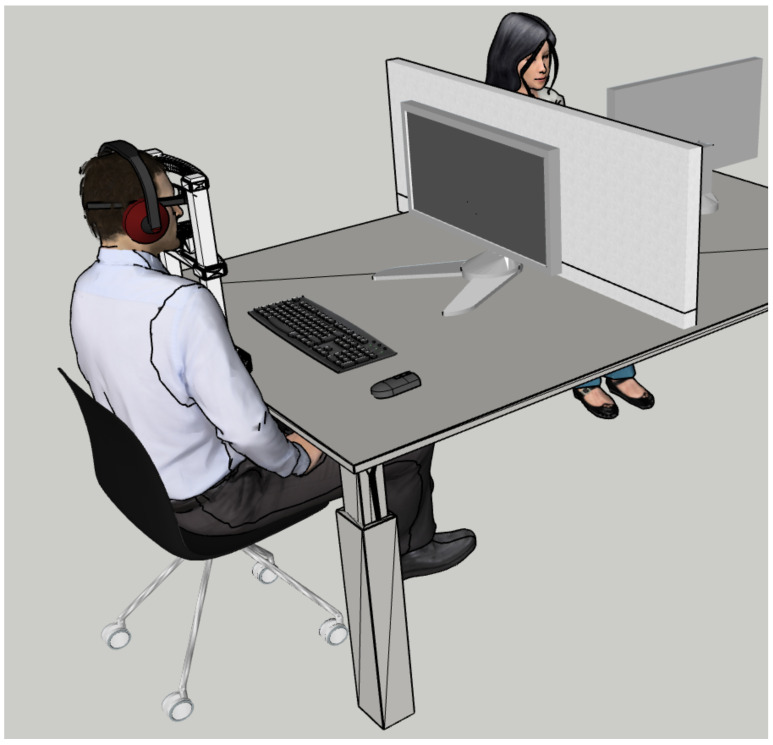
Graphical illustration of the experimental apparatus.

**Figure 3 brainsci-13-00589-f003:**
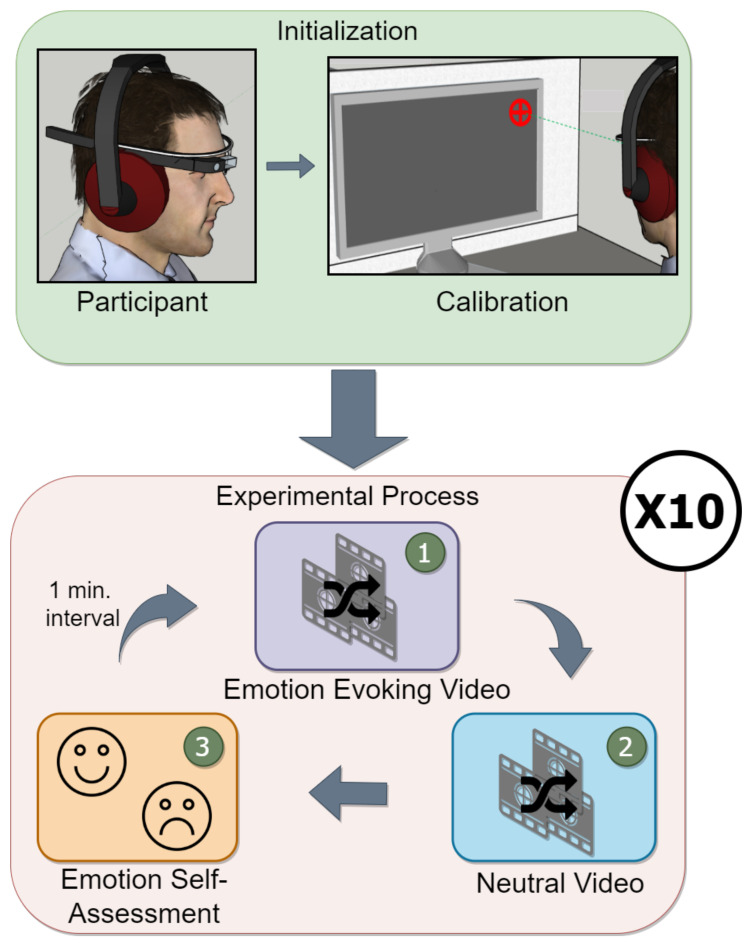
Design of the experimental study.

**Figure 4 brainsci-13-00589-f004:**
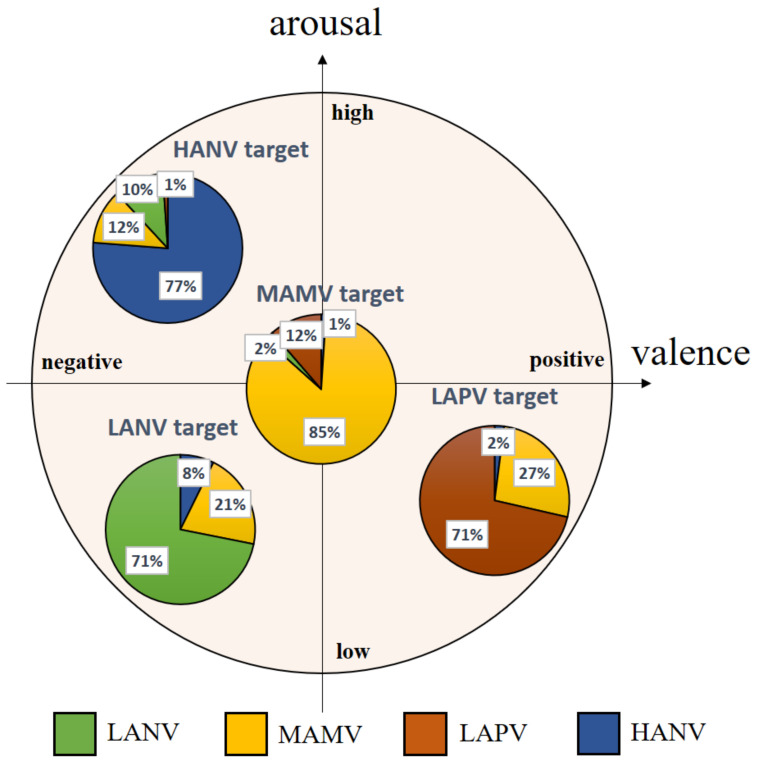
Subjective annotation in reference to objective annotation, in the valence–arousal space. HANV: High Arousal–Negative Valence, LANV: Low Arousal–Negative Valence, LAPV: Low Arousal–Positive Valence, MAMV: Medium Arousal–Medium Valence.

**Figure 5 brainsci-13-00589-f005:**
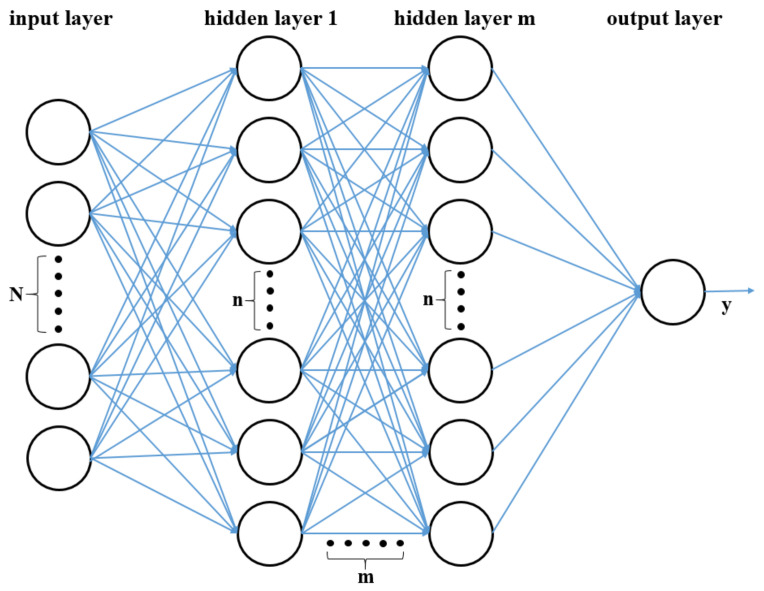
Architecture of the implemented neural networks.

**Table 1 brainsci-13-00589-t001:** Emotion-evoking videos.

Video	Duration (min)	Emotion
American history X	1.17	Anger
Benny and Joon	2.01	Tenderness
Blue (2)	0.40	Neutral
Ghost	1.19	Tenderness
Man bites dog (1)	1.32	Disgust
Schindler’s list (1)	0.58	Sadness
Schindler’s list (2)	1.55	Anger
The dreamlife of angels	2.00	Sadness
The lover	0.43	Neutral
Trainspotting (2)	1.00	Disgust

The FilmStim database contains several video clips that belong to the same movie. The digits in parenthesis in the first column refer to the video clip number belonging to the respective movie.

**Table 2 brainsci-13-00589-t002:** Estimated mean values ± standard errors of all features for three classes of EA and EV.

		EA			EV	
	LA	MA	HA	NV	MV	PV
Feature	Mean	Mean	Mean	Mean	Mean	Mean
Fix. freq. (fix./s)	2.10 ± 0.06	2.09 ± 0.06	2.20 ± 0.06	2.17 ± 0.06	2.09 ± 0.06	2.09 ± 0.07
Fix. duration (ms)	314.50 ± 8.65	306.94 ± 8.90	295.52 ± 8.72	303.62 ± 8.33	307.48 ± 8.89	310.83 ± 9.59
Variation	0.87 ± 0.02	0.89 ± 0.02	0.87 ± 0.02	0.87 ± 0.02	0.89 ± 0.02	0.89 ± 0.02
Skewness	2.51 ± 0.08	2.38 ± 0.08	2.64 ± 0.08	2.58 ± 0.07	2.38 ± 0.08	2.56 ± 0.10
Kurtosis	9.48 ± 0.68	7.92 ± 0.73	10.92 ± 0.71	10.35 ± 0.59	7.91 ± 0.73	9.59 ± 0.93
Sac. freq. (sac./s)	2.48 ± 0.18	2.70 ± 0.19	3.05 ± 0.18	2.83 ± 0.17	2.68 ± 0.19	2.54 ± 0.21
Sac. amplitude (deg.)	14.01 ± 0.14	13.88 ± 0.14	14.00 ± 0.14	14.08 ± 0.13	13.89 ± 0.14	13.77 ± 0.15
Variation	0.18 ± 0.01	0.20 ± 0.01	0.18 ± 0.01	0.18 ± 0.01	0.20 ± 0.01	0.18 ± 0.01
Skewness	0.23 ± 0.20	0.37 ± 0.21	0.34 ± 0.21	0.24 ± 0.20	0.36 ± 0.21	0.42 ± 0.23
Kurtosis	7.44 ± 0.68	6.59 ± 0.73	7.85 ± 0.70	7.60 ± 0.62	6.58 ± 0.73	7.76 ± 0.86
Sac. velocity (°/s)	223.90 ± 11.07	240.77 ± 11.34	234.27 ± 11.15	231.71 ± 10.76	240.63 ± 11.34	220.16 ± 12.05
Variation	0.78 ± 0.02	0.74 ± 0.02	0.74 ± 0.02	0.76 ± 0.02	0.74 ± 0.02	0.76 ± 0.02
Skewness	1.35 ± 0.15	1.08 ± 0.15	1.20 ± 0.15	1.22 ± 0.14	1.08 ± 0.15	1.47 ± 0.16
Kurtosis	2.43 ± 0.81	1.41 ± 0.85	2.21 ± 0.82	2.05 ± 0.76	1.40 ± 0.85	3.18 ± 0.95
Peak velocity (°/s)	299.83 ± 13.31	320.39 ± 13.60	310.35 ± 13.39	308.00 ± 12.97	320.27 ± 13.60	295.27 ± 14.38
Variation	0.75 ± 0.02	0.72 ± 0.02	0.71 ± 0.02	0.73 ± 0.02	0.72 ± 0.02	0.74 ± 0.02
Skewness	1.00 ± 0.14	0.78 ± 0.15	0.87 ± 0.14	0.89 ± 0.14	0.78 ± 0.15	1.08 ± 0.15
Kurtosis	1.10 ± 0.56	0.59 ± 0.58	1.03 ± 0.57	0.92 ± 0.54	0.58 ± 0.58	1.54 ± 0.64
Sac. duration (ms)	22.03 ± 0.46	23.63 ± 0.47	22.63 ± 0.46	22.46 ± 0.43	23.63 ± 0.47	21.87 ± 0.52
Variation	0.99 ± 0.03	0.96 ± 0.03	1.04 ± 0.03	1.02 ± 0.03	0.96 ± 0.03	0.97 ± 0.04
Skewness	2.73 ± 0.11	2.60 ± 0.11	2.96 ± 0.11	2.87 ± 0.01	2.60 ± 0.11	2.74 ± 0.13
Kurtosis	9.87 ± 0.91	9.55 ± 0.97	12.21 ± 0.93	11.29 ± 0.83	9.51 ± 0.97	10.05 ± 1.16
Blink freq. (blinks/s)	0.21 ± 0.02	0.27 ± 0.03	0.22 ± 0.03	0.23 ± 0.02	0.27 ± 0.03	0.19 ± 0.03
Blink duration (ms)	215.32 ± 4.06	214.88 ± 4.17	220.45 ± 4.09	218.83 ± 3.91	214.79 ± 4.16	214.61 ± 4.48
Pupil diameter (mm)	3.85 ± 0.10	3.74 ± 0.10	3.95 ± 0.10	3.87 ± 0.10	3.74 ± 0.10	4.00 ± 0.11
Variation	0.067 ± 0.003	0.076 ± 0.004	0.076 ± 0.003	0.073 ± 0.003	0.076 ± 0.004	0.067 ± 0.004
Skewness	−0.29 ± 0.07	−0.28 ± 0.08	−0.29 ± 0.07	−0.23 ± 0.06	−0.27 ± 0.08	−0.47 ± 0.10
Kurtosis	2.34 ± 0.34	1.71 ± 0.36	1.86 ± 0.35	1.80 ± 0.30	1.71 ± 0.36	3.09 ± 0.44

Fixation duration, saccade amplitude and saccade duration are median values. Saccade velocity, peak saccade velocity, blink duration and pupil diameter are mean values.

**Table 3 brainsci-13-00589-t003:** Questionnaire scores.

Questionnaire	Mean	St. dev.	Min	Max
CES-D	10.4	4.4	3	19
Cognitive empathy	3.7	0.4	2.9	5.0
Emotional empathy	3.3	0.6	1.6	4.8
State anxiety	53.3	6.2	36	74
Trait anxiety	54.9	6.9	42	70

**Table 4 brainsci-13-00589-t004:** Results for target emotions.

Target Emotion	Hit Rate (%)	Mean Rating for Target Emotion
Anger	53	5.9
Disgust	79	7.4
Sadness	71	6.0
Tenderness	71	5.1
Neutral	85	-

**Table 5 brainsci-13-00589-t005:** Emotion ratings for each subjective emotion annotation.

Annotation	Anger	Disgust	Sadness	Tenderness
Anger	7.7	4.5	5.7	0.2
Disgust	4.2	8.4	4.0	0.1
Sadness	3.8	3.3	7.5	0.1
Tenderness	0.4	0.1	0.1	6.4

**Table 6 brainsci-13-00589-t006:** Results of Mixed Linear Model (MLM) and post hoc tests for arousal levels.

	MLM	Post Hoc		
Feature	*p*	LA-MA	LA-HA	MA-HA
Fixation frequency	**0.009**	1.000	**0.030**	**0.027**
Fixation duration	**0.005**	0.744	**0.003**	0.230
Fixation dur. Sk.	**0.043**	0.700	0.478	**0.038**
Fixation dur. Kurt.	**0.007**	0.290	0.328	**0.005**
Saccade frequency	**0.001**	0.604	**0.000**	0.095
Saccade amp. Var.	**0.043**	**0.048**	1.000	0.139
Saccade vel. Var	**0.014**	0.065	**0.023**	1.000
Saccade vel. Sk.	**0.045**	**0.043**	0.367	0.818
Peak sac. vel. Var.	**0.019**	0.132	**0.023**	1.000
Saccade dur.	**0.000**	**0.000**	0.257	**0.029**
Saccade dur. Var	**0.004**	0.597	0.089	**0.004**
Saccade dur. Sk.	**0.005**	0.830	0.076	**0.005**
Saccade dur. Kurt.	**0.021**	1.000	0.060	**0.045**
Blink frequency	**0.001**	**0.001**	1.000	**0.004**
Pupil diameter	**0.000**	0.08	0.093	**0.000**
Pupil diam. Var.	**0.008**	**0.042**	**0.014**	1.000

Statistically significant differences (*p* < 0.05) are in bold. LA: low arousal, MA: medium arousal, HA: high arousal, Sk: skewness, Kurt: kurtosis.

**Table 7 brainsci-13-00589-t007:** Results of Mixed Linear Model (MLM) and post hoc tests for valence levels.

Feature	*p*	NV-MV	NV-PV	MV-PV
Fixation frequency	**0.045**	0.120	0.187	1.000
Fixation dur. Kurt.	**0.020**	**0.015**	1.000	0.419
Saccade amplitude	**0.000**	**0.047**	**0.001**	0.701
Saccade amp. Var	**0.047**	0.063	1.000	0.128
Saccade vel. Sk	**0.011**	0.535	0.065	**0.009**
Peak sac.vel. Sk.	**0.023**	0.619	0.116	**0.019**
Saccade duration	**0.000**	**0.004**	0.400	**0.001**
Saccade dur. Var	**0.003**	**0.007**	0.074	1.000
Saccade dur. Sk.	**0.031**	**0.029**	0.751	0.950
Blink frequency	**0.000**	**0.005**	0.068	**0.000**
Pupil diameter	**0.000**	**0.013**	**0.018**	**0.000**
Pupil diam. Kurt.	**0.010**	1.000	**0.013**	**0.021**

Statistically significant differences (*p* < 0.05) are in bold. NV: negative valence, MV: medium valence, PV: positive valence, Sk: skewness, Kurt: kurtosis.

**Table 8 brainsci-13-00589-t008:** Results of Mixed Linear Model (MLM) and post hoc tests for emotion classes.

	MLM	Bonferroni post hoc					
Feature	*p*	1	2	3	4	5	6
Fixation frequency	**0.019**	1.000	1.000	0.483	1.000	**0.046**	0.086
Fixation duration	**0.012**	1.000	1.000	**0.014**	1.000	0.353	0.176
Fixation dur. Kurt.	**0.019**	0.819	1.000	1.000	1.000	**0.010**	1.000
Saccade frequency	**0.002**	1.000	1.000	**0.003**	1.000	0.182	**0.034**
Saccade amplitude	**0.000**	**0.005**	**0.000**	0.078	1.000	1.000	0.081
Saccade velocity Var	**0.017**	0.052	1.000	**0.022**	1.000	1.000	0.820
Saccade velocity Sk.	**0.021**	0.918	0.559	1.000	**0.012**	1.000	0.133
Peak saccade velocity Var.	**0.019**	0.083	1.000	**0.017**	1.000	1.000	0.907
Peak saccade velocity Sk.	**0.041**	0.849	1.000	1.000	**0.030**	1.000	0.208
Saccade duration	**0.001**	**0.008**	1.000	1.000	**0.001**	0.059	0.451
Saccade duration Var.	**0.004**	0.440	1.000	1.000	1.000	**0.007**	0.068
Saccade duration Sk.	**0.015**	1.000	1.000	0.279	1.000	**0.012**	0.496
Blink frequency	**0.000**	0.299	0.074	1.000	**0.000**	**0.007**	0.352
Pupil diameter	**0.000**	1.000	**0.000**	**0.000**	**0.000**	**0.000**	1.000
Pupil diameter Var.	**0.025**	0.212	1.000	0.102	0.297	1.000	0.148
Pupil diameter Sk.	**0.037**	0.961	**0.022**	0.680	0.470	1.000	0.540
Pupil diameter Kurt.	**0.020**	1.000	**0.043**	1.000	**0.036**	1.000	0.064

Statistically significant differences (*p* < 0.05) are in bold. 1 = LANV-MAMV, 2 = LANV-LAPV, 3 = LANV-HANV, 4 = MAMV-LAPV, 5 = MAMV-HANV, 6 = LAPV-HANV. HANV: High Arousal–Negative Valence, LANV: Low Arousal–Negative Valence, LAPV: Low Arousal–Positive Valence, MAMV: Medium Arousal–Medium Valence. Var: variation Sk: skewness, Kurt: kurtosis.

**Table 9 brainsci-13-00589-t009:** EA classification results.

Classes	Hidden Layer Size	AUC	f1-Score	Acc.
LA + MA/HA	128, 32	0.73	0.74	0.74
HA/LA	128, 64	0.66	0.67	0.67
HA/MA	128, 64	0.74	0.73	0.75
MA/LA	128, 64	0.80	0.79	0.81
HA/MA/LA	256, 64, 32	0.72	0.73	0.74

**Table 10 brainsci-13-00589-t010:** EV classification results.

Classes	Hidden Layer Size	AUC	f1-Score	Acc.
NV + MV/PV	64, 16	0.91	0.91	0.92
PV/NV	32, 8	0.82	0.82	0.82
PV/MV	32, 8	0.87	0.89	0.90
MV/NV	64, 32	0.58	0.57	0.59
PV/MV/NV	128, 64, 32	0.84	0.84	0.85

**Table 11 brainsci-13-00589-t011:** Synchronous EA and EV classification results.

Classes	Hidden Layer Size	AUC	f1-Score	Acc.
MAMV/HANV/LANV/LAPV	128, 64, 16	0.73	0.76	0.72

## Data Availability

The resulting eSEE-d is available to the academic community (DOI: 10.5281/zenodo.5775674).

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
