# Peer review of "eSEE-d: Emotional State Estimation Based on Eye-Tracking Dataset"

_brainsci, 2023, doi:10.3390/brainsci13040589_

Round 1

Reviewer 1 Report

The authors developed a novel emotional experimental paradigm for recording Eye-tracking data. The description of the paradigm is detailed, and the analysis results are acceptable. Here are some issues should be addressed.

Section 1:

1) In emotion recognition research, the construction of emotional databases is important, and emotional databases might be connected with different modality signals, different stimulus materials, and different experimental paradigms. Thus, considering that the authors proposed an emotional dataset, the authors should give a brief review of such emotional databases, and clarify the motivation. In order to enhance the scientific rigor of the study, it is recommended to the following papers, doi:10.1109/TIM.2022.3149116, doi: 10.1109/TNSRE.2023.3253866. 

Section 2:

2) The authors have chosen five emotions (anger, disgust, sadness, tenderness, neutral), and labeled them as High Arousal - Negative Valence (HANV), Low Arousal - Negative Valence (LANV), Low Arousal - Positive Valence (LAPV), and Medium Arousal - Medium Valence (MAMV). However, why didn’t the authors choose those emotions that could be labeled as High Arousal - Positive Valence (HAPV), such as happiness?

3) In table 1, what’s the mean of the digit of the brackets in the first column represent?  

Section 3:

4) Please give a more concrete description and reference of the Mixed Linear Model analysis.

Section 4: 

5) The authors should state a future work in order to discuss the feature selection method for authors only employed the statistics method to choose the eye tracking features in this work.

Reviewer 2 Report

In the paper, a dataset based on eye-tracking data is presented to estimate emotional states.  

- There is an acceptable number of participants.In my opinion, the main contribution of the paper is the number of participants.A serious problem with the dataset is that videos are shorter than other datasets, which completely impacts its quality.

- According to the authors, the dataset has more subjects, which enables researchers to apply modern artificial intelligence. Researchers can't apply modern AI to eye-tracking data collected in brief trials.

- My surprise when I downloaded the videos was that some were black and white and others were colored. There are 10 videos, and this difference could affect the results. If there is another paper that does the same thing, please cite it. No papers were found.

- In terms of the number of subjects, duration of videos, number of videos or pictures, and quality of videos, compare the dataset with others.

- Is there a reason why the authors didn't show images to the participant? A complete explanation of the benefits of showing videos to subjects should be given by the authors.

- Explain why the authors believe there is enough data from different emotional states in the dataset.

- There is no logical introduction to the background of the field, no analysis of the current problems in another dataset, and no explanation of why new data should be collected.

- Why is the work being done? There is no sound motivation for the problem considered. It is essential that the authors demonstrate a clear scientific interest in the objectives and results of the study.

- There is a need for the authors to revise the main objectives of the present datasets. Real objectives should be mentioned.

Round 2

Reviewer 2 Report

As a result of the author's corrections, it appears to be acceptable.